# Surgical Strategy to Decrease the Revision Rate of Fassier–Duval Nailing in the Lower Limbs of Osteogenesis Imperfecta

**DOI:** 10.3390/jpm12071151

**Published:** 2022-07-15

**Authors:** Yi-Chi Hung, Kai-Yuan Cheng, Hsiang-Yu Lin, Shuan-Pei Lin, Chen-Yu Yang, Shih-Chia Liu

**Affiliations:** 1Department of Orthopedics, MacKay Memorial Hospital, Taipei 10449, Taiwan; tina.5210@mmh.org.tw (Y.-C.H.); ai3keiko@gmail.com (K.-Y.C.); jasonscliu649@gmail.com (S.-C.L.); 2Department of Pediatrics, MacKay Memorial Hospital, Taipei 10449, Taiwan; linhy@mmh.org.tw (H.-Y.L.); rarediseasemmh@gmail.com (S.-P.L.); 3Department of Rare Disease Center, MacKay Memorial Hospital, Taipei 10449, Taiwan; 4Department of Medicine, MacKay Medical College, New Taipei City 25245, Taiwan; 5MacKay Junior College of Medicine, Nursing and Management, Taipei 11260, Taiwan; 6Department of Medical Research, China Medical University Hospital, China Medical University, Taichung 406040, Taiwan; 7Department of Infant and Child Care, National Taipei University of Nursing and Health Sciences, Taipei 10650, Taiwan

**Keywords:** osteogenesis imperfecta, Fassier–Duval, telescoping, revision, migration, complication, surgical strategy, Sillence

## Abstract

(1) Background: The Fassier–Duval (FD) nail was developed for the treatment of osteogenesis imperfecta (OI). The aim of this study was to review the results of OI patients treated with the FD nail at our institution and discuss a surgical strategy to decrease the FD nail revision rate; (2) Methods: We retrospectively reviewed OI patients treated at our institution between 2015 and 2020. OI patients treated with FD nail insertion in the long bones of the lower extremities were included, and those with a follow-up duration <1 year or incomplete radiographs were excluded. Data on the type of OI, age, sex, use of bisphosphonate treatment, and nail failure were recorded; (3) Results: The final cohort consisted of seven patients (three females and four males) with ten femurs and ten tibiae involved. Six of the patients had type III OI, and one had type IV OI. An exchange of implant was required in 11 limbs. The average interval between previous FD nail insertion and revision surgery was 2.4 years; (4) Discussion: The main reasons for revision surgery were migration of the male/female component, refracture/nail bending, and delayed union. In the femur, migration of the female component or nail bending were common reasons for failure, while migration of the male component and delayed union were common in the tibia; (5) Conclusions: Surgery for OI patients is challenging, and physicians should aim to minimize complications and the need for revision. Sufficient depth of purchase, center–center nail position, and adequate osteotomy to correct bowing are the key factors when using the FD nail.

## 1. Introduction

Osteogenesis imperfecta (OI) is a rare genetic collagen disease characterized by bone fragility and ligamentous laxity [1,2]. Surgical fixation of brittle bones is difficult and is associated with a high rate of complications. The Fassier–Duval (FD) telescoping nail was developed to provide stability during long bone growth, and, currently, it is the most commonly used intramedullary device worldwide [3,4,5]. However, the surgical revision rate is still high [4,6,7]. The aim of this study was to review the results of OI patients treated with the FD nail at our institution and discuss a surgical strategy to decrease the FD nail revision rate.

## 2. Materials and Methods

We retrospectively reviewed all patients with a diagnosis of OI who were treated at our institution between 2015 and 2020. OI patients with at least one FD nail implant procedure performed in one lower extremity long bone were included. The exclusion criteria were those with a follow-up duration less than one year, incomplete radiograph images, and an FD nail implanted in an upper extremity long bone. The following data were collected: age at first surgery, sex, Sillence classification of OI, usage of bisphosphonate treatment, location of nail implant, nail diameter, revision timing, and failure mode.

The Sillence classification, defined as type I to type IV based on clinical finding, is currently most widely referenced [8]. The phenotype of OI in our study was proven by molecular genetic examinations [1,9,10] and confirmed by our pediatric geneticists.

With regards to the FD nail technique, the infrapatellar approach was used to apply tibial FD nails, while the antegrade approach was used for femur nails. For the tibial FD nails, a short threaded male component without cross-pin fixation was chosen for distal fixation, while a short or long threaded nail was chosen according to the thickness of the epiphysis in the distal femur. The FD nails were available in five diameters: 3.2, 4.0, 4.8, 5.6, and 6.4 mm [11].

Serial X-ray images were examined for bone healing and implant failure. Nail failure was defined as: (1) migration of the male component (distally dislodged; the male nail tip had been pulled out of the epiphysis); (2) migration of the female component (proximal protrusion out of the anchorage; the female component had become dislodged); (3) refracture of the bone; (4) nail bending; and (5) delayed union. Delayed union was defined as failure to reach bony union by 6 months.

## 3. Results

The final cohort consisted of seven patients (three females and four males) with twenty lower limb long bones involved (ten femurs and ten tibiae). According to Sillence classification, six patients were classified as having type III OI, and one patient with type IV OI. The average age at first surgery was 5.4 years (range 1.5–11 years). The average nail diameter was 4.2 mm (range 3.2–6.4 mm). Among the 20 affected limbs, 11 required revision surgery, for a revision rate of 55%. The average interval between previous FD nail insertion and revision surgery was 2.4 years (range 0.8–3.25 years). Patient demographics and mode of nail failure are described in Table 1. All patients received regular intravenous bisphosphonate treatment with pamidronate disodium at a dose of 30 mg/m^2^, once every month. The treatment was stopped after surgery but resumed post-operatively after 3 months.

Among the causes of FD nail failure, migration of the male component (femur: 1, tibia: 4), migration of the female component (femur: 3), nail bending (femur: 3), refracture of the bone (femur: 2), and delayed union (tibia: 3) are described in Table 2.

## 4. Discussion

Osteogenesis imperfecta (OI) is a rare genetic collagen disease characterized by bone fragility and ligamentous laxity [1,2]. Typical surgical interventions include realignment osteotomies and intramedullary fixation. However, internal fixation with nontelescoping intramedullary nails requires frequent revision due to patient growth [3]. Telescoping intramedullary nails have superior probability of survival and decreased revision rate compared with static implants [12,13,14]. Since 2001, the FD telescopic system has been used in the treatment of OI, congenital pseudarthrosis of the tibia, and, also, other various skeletal dysplasias [3,4,5]. The FD telescopic nail has the advantage over regular nails [15,16] of being inserted through one incision without arthrotomy [14,17], however, complications can still occur, with a complication rate ranging from 0% to 53% [4,6,7].

In a review of the literature, the main reasons for revision surgery were migration of the male/female component, refracture/nail bending, and delayed union [7,18,19]. In our series, migration of the female component and nail bending were common reasons for failure in the femur, while migration of the male component and delayed union were common reasons for failure in the tibia, which is described in Table 2.

Musielak, B.J. et al. [7] reported that the risk factors for FD nail failure were a patient age younger than 5 years and an FD nail diameter smaller than 4 mm. However, few studies have described the possible reasons for nail failure in OI patients. According to Holmes et al. [20], one of the most important reasons leading to implant migration is the position of the distal end of the nail in relation to the center of the epiphysis. They reported a 12% increase in failure rate per 10% increase in deviation from the center for both anteroposterior and lateral radiographs.

However, we still experienced failures where a nail in a fairly central position became dislodged or the bone refractured. Therefore, we suggest that not only the position of the nail but also the depth of the purchase and the overall mechanical axis alignment are important, along with a central nail position and bowing correction. According to our failure cases, the main principles for the FD nail should be purchase, nail position, and bowing.

### 4.1. Principle 1: Enough Depth of Nail Purchase in Distal and Proximal Epiphysis

In a case of a 4-year-old female with type III OI (Figure 1), the position of the male component was quite central in the distal epiphysis on both anteroposterior and lateral views. However, there was insufficient purchase of the male component tip (Figure 1a). If the threads end just around the physis instead of beyond it (Figure 1b), distal dislodgement would occur along with growth of the limb (Figure 1c). However, this raises the question of how to ensure sufficient depth of the male implant intraoperatively. We suggest the following steps: First, center the male component at the center of the distal epiphysis as far as possible so that the thickest epiphysis is purchased. The thread of the male component needs to be fully submerged into the epiphysis, and the flange of the nail must exceed the physis, or at least stop at the level of the physis. If the distal epiphysis of the tibia is small, the tip of the nail may reach the subchondral area to ensure that the whole thread goes through the physis (Figure 2). Since the distal epiphysis of the femur is larger than the distal epiphysis of the tibia, there is a relatively lower risk of distal dislodgement in the femur (tibia 4, femur 1 in our series) due to sufficient distal purchase.

Migration of the male component is common in the tibia due to an almost empty epiphysis [18,19]. For the male component, short thread screws are designed to anchor the epiphysis and resist a pulling out force due to growth [15]. A keyhole over the screw thread adds strength to the distal fixation. The fixation-wire with a diameter ranging from 0.7 mm to 1.1 mm for the corresponding nail can increase more anchorage [11], however, it was relatively difficult to facilitate targeting the keyhole with such a small fixation-wire. In addition, non-threaded (LON) male components with peg fixation and larger keyhole (1.5 mm) [11] are not currently available in Taiwan. Thus, we could only follow the principles mentioned above to ensure maximal distal fixation.

Moreover, loss of fixation in the proximal femur is also related to insufficient purchase, which can be subdivided into two situations: (1) the nail entry point is not precisely over the tip of the greater trochanter and (2) insufficient depth of the thread of the female component. Proximal femoral fixation is challenging in younger children due to a partially ossified greater trochanter [18].

Another of our cases involved a boy who received double osteotomies and fixation with an FD nail in his right femur at the age of 1.5 years (Figure 3). The proximal osteotomy segment was too small to align and the greater trochanter had not ossified, so it was difficult to find the most suitable entry point under the c-arm (Figure 3a). During follow-up, we found that the female thread had become much more distal and anterior (Figure 3b). In the following revision, due to concerns over damaging the apophysis of the femur and iatrogenic greater trochanter epiphysiodesis, with resulting sequela of valgus deformity, insufficient depth of the female component may have occurred, resulting in its proximal protrusion (Figure 3c,d). The importance of the entry point and adequate purchase of the proximal femur cannot be over-emphasized (Figure 3e).

### 4.2. Principle 2: The Nail Must Be Placed in the Center of the Epiphysis on Both Anteroposterior and Lateral Views

An 11-year-old boy with type IV OI (Figure 4) underwent revision surgery with an FD nail due to implant malposition. When the distal tip of a nail is not placed in the center of the joint, the mechanical axis may become deviated, possibly leading to refracture or nail bending (Figure 4a). In this patient, deviated alignment and a bowed femur caused an imbalance in mechanical stress, which led to a proximal femur refracture. Restoration of the alignment and bowing correction to avoid further implant failure were performed with a pediatric nail after physeal closure (Figure 4b).

First, c-arm positioning intra-operatively cannot be over-emphasized to obtain the true AP and true lateral view of the affected limb. We prefer using the supine position or fracture table during both tibia and femur interventions, thus changing the projection radiation instead of changing the limb position. It is important to try to identify the entry point as precisely as possible. This is especially difficult for proximal femur entry because of the non-ossified small greater trochanter in young children under a non-open and percutaneous approach. The open approach is a possible solution; however, more tissue damage is a concern. Thus, we still use the percutaneous approach under fluoroscopy and accept the possible deviation rather than risk possible damage from the open approach. After a straight and rigid guide-wire or reamer has been inserted from the correct entry point, the wire will eventually meet the area of deformity, or CORA. We use a multiple percutaneous drilling technique for osteoclasis at the CORA and straighten the limb as the guidewire passes through, which realigns the deformed bone and minimizes surgical trauma.

### 4.3. Principle 3: Bowing of the Bone Is Related to Delayed Union and Refracture, and Correction Is Warranted

A 4-year-old male with type III OI presented as a classic model of insufficient depth of purchase and excessive bowing of the tibia (Figure 5). When axial force is applied to the limbs with excessive bowing, there will be an uneven force on the concave and convex sides, leading to delayed union (Figure 5a–c). Correction of the bowing with adequate osteotomy (or osteoclasis) before FD nail insertion is crucial (Figure 5d,e).

These principles may help to guide treatment and avoid complications. Six years after surgery, a male patient with type III OI (Figure 6) still maintained good nail telescoping without implant failure. We emphasize that sufficient depth of purchase is important to avoid migration of both the male and female components of the FD nail. A central nail position in the epiphysis ensures balanced alignment of the mechanical axis. Correcting excessive bowing cannot be over-emphasized. Following these principles, it is possible to avoid most implant-related complications during the long course of treatment for OI patients.

All patients received regular intravenous bisphosphonate treatment with pamidronate disodium at a dose of 30 mg/m^2^, once every month. The treatment was stopped after surgery but was resumed post-operatively after 3 months. Pamidronate is a bisphosphonate, and it has been shown to increase bone mineral density (BMD) and decrease fracture rate, thereby substantially improving functional status [21,22]. During the first pamidronate infusion, a few patients had low-grade fever. These patients received acetaminophen prior to subsequent pamidronate infusions, and the fever did not recur. No other associated adverse effects were observed.

## 5. Limitation

Osteogenesis imperfecta is a rare disease, and one of the limitations of this study is the small number of cases treated at a single center with two pediatric orthopedic surgeons. A larger number of patients from multiple centers with a longer follow-up duration in future studies will be greatly beneficial to clarify the treatment of choice. This surgical strategy could be considered as a proposed study method in further collaborations with other medical institutes for a larger number of cases and, possibly, as a prospective series. Optimal and evidence-based treatments are still being studied. In addition, wire fixation of the male component in the distal tibia could not be performed, as this implant type is not currently available in Taiwan, which may have led to a higher dislodgment rate in the distal tibia. Further studies are warranted to investigate the effect of implant selection on the revision rate.

## 6. Conclusions

Surgery for OI patients is challenging. Orthopedic surgeons should aim to minimize complication and revision rates with the FD nail. In the femur, migration of the female component and nail bending were common reasons for failure in our series, while migration of the male component and delayed union were common in the tibia. According to our experience of using the FD telescoping nail, the key factors are sufficient depth of purchase, center–center nail position, and adequate osteotomy to correct bowing.

## Figures and Tables

**Figure 1 jpm-12-01151-f001:**
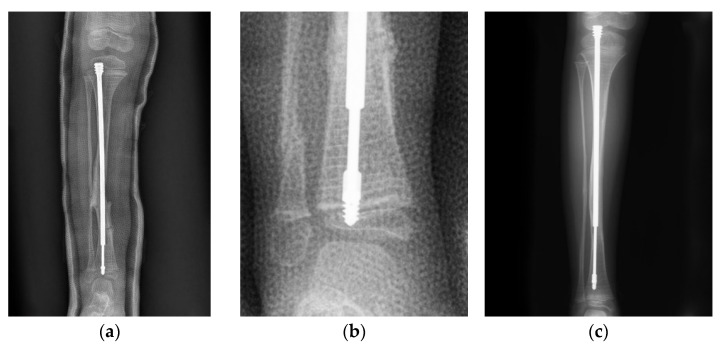
Anteroposterior (AP) view radiograph of the lower leg of a 4-year-old female post FD nail implantation. (**a**) The position of the nail was quite central in the distal epiphysis. (**b**) Looking closely, there was insufficient purchase of the nail. (**c**) As the patient grew, the male component migrated (distal tibial nail pulled out of the epiphysis) 1.5 years after implantation.

**Figure 2 jpm-12-01151-f002:**
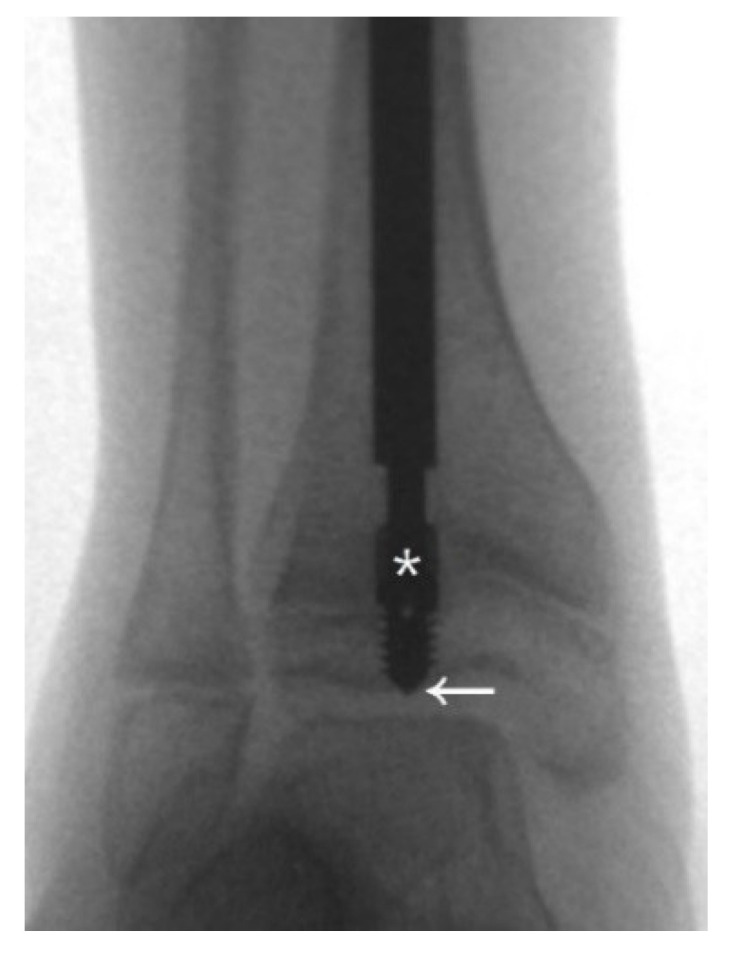
The first principle to decrease the revision rate is to ensure sufficient purchase. The thread of the male component needs to be fully submerged into the epiphysis, and the flange (*) of the nail must exceed the physis or at least stop at the level of the physis. If the distal epiphysis of the tibia is small, the tip of the nail may reach the subchondral area (arrow) to ensure that the whole thread goes through the physis.

**Figure 3 jpm-12-01151-f003:**
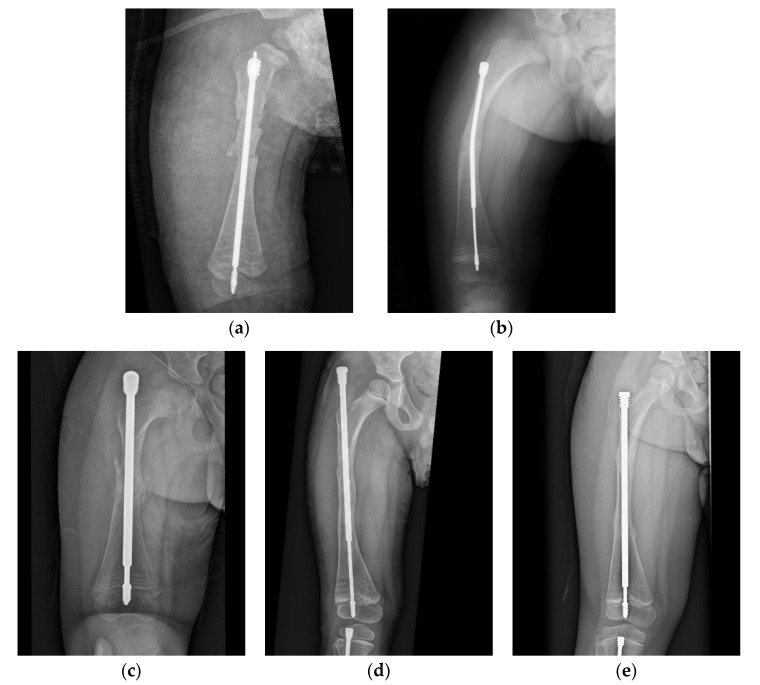
Anteroposterior (AP) view radiograph of the femur of a 1.5-year-old male post FD nail implantation. (**a**) The boy received double osteotomies and fixation with an FD nail in the right femur. The proximal fragment was extremely small and difficult to control, and the greater trochanter had not yet been ossified, so it was a huge challenge to find the most suitable entry point under the c-arm. We can see the entry point was anterior to the true position of the greater trochanter. (**b**) During regular follow-up, we found that the female thread had become much more distal and anterior with nail bending. (**c**) In the revision surgery, due to concerns over damaging the apophysis of the femur in order to avoid physis fusion and valgus deformity, we may have used insufficient depth of the female component. (**d**) During regular follow-up, we found proximal protrusion of the female component. (**e**) Good position and adequate purchase of the FD nail post revision surgery.

**Figure 4 jpm-12-01151-f004:**
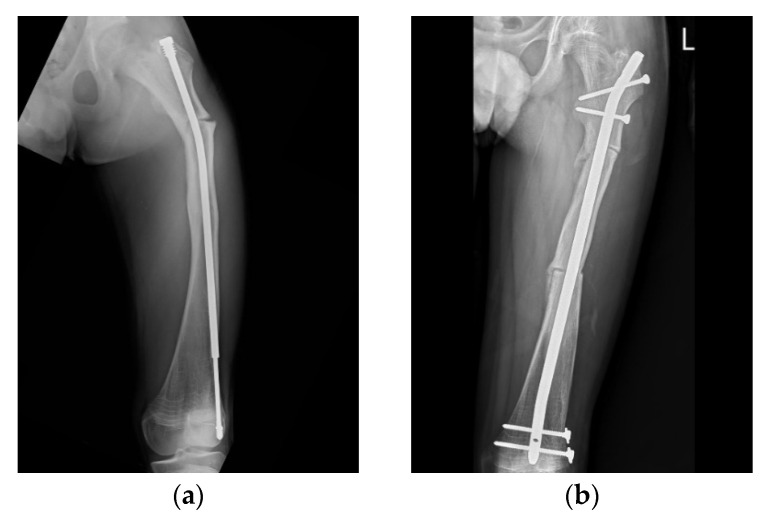
Anteroposterior (AP) radiograph of the femur of an 11-year-old male post FD nail implantation. (**a**) The position of the male component was eccentric in the distal epiphysis. Due to an incorrect mechanical axis, subsequent refracture with nail bending was observed 3 years after surgery. (**b**) Revision surgery of osteoclasis and fixation with an interlocking PediNail^TM^.

**Figure 5 jpm-12-01151-f005:**
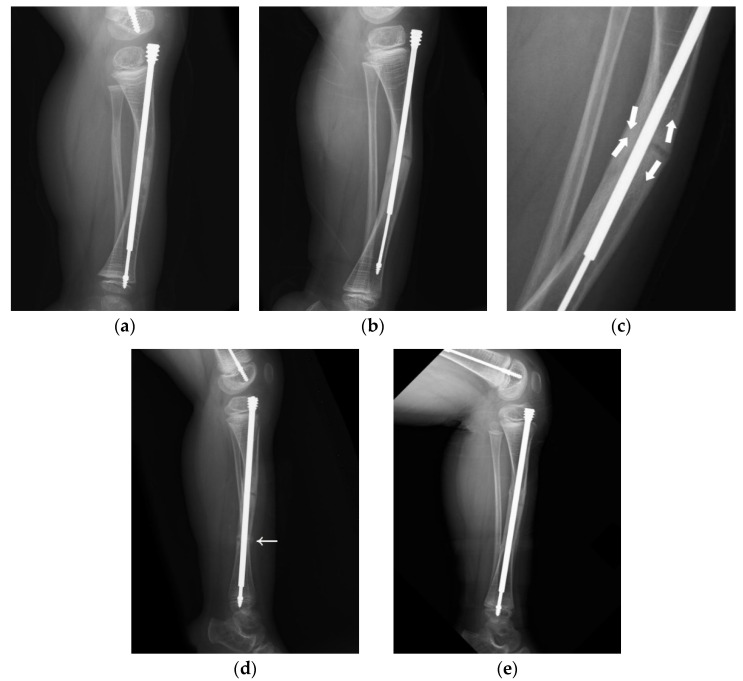
Lateral radiograph of the lower leg of a 4-year-old male post FD nail implantation. (**a**) There was insufficient depth of purchase into the epiphysis, and the position of the nail was anterior to the center of the epiphysis due to excessive bowing of the tibia. (**b**) As the patient grew, migration of the male implant (distal tibial nail pulled out of the epiphysis) was noted 2.5 years after implantation. (**c**) Excessive bowing resulted in uneven force applied to the concave and convex sides, which caused thickening of the cortex on the concave side and delayed union on the convex side. The aim of revision surgery was to correct bowing of the tibia with sufficient depth of purchase. (**d**) We corrected the bowing with adequate osteoclasis (arrow) before FD nail insertion. The position of the nail was at the center of the epiphysis in the lateral view after correcting for excessive bowing. (**e**) Good position of the FD nail and bone healing during regular follow-up.

**Figure 6 jpm-12-01151-f006:**
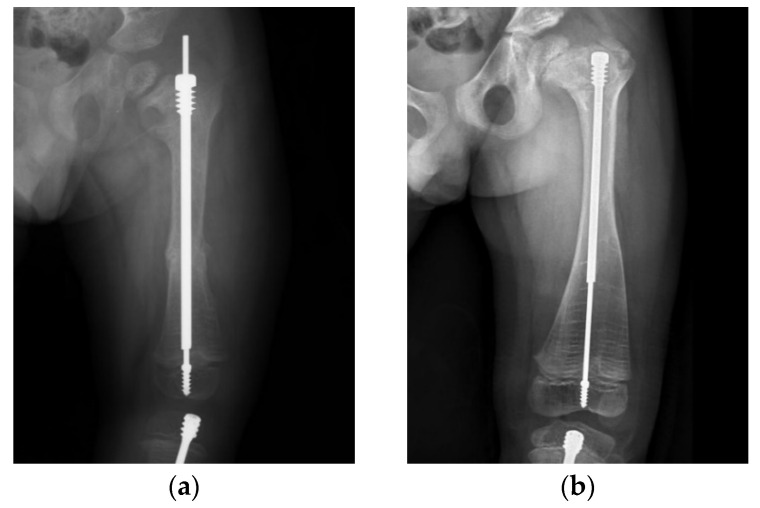
Anteroposterior (AP) radiograph of the left femur in a male patient with type III OI post FD nail implantation. (**a**) Initial radiograph post FD nail implantation at 4 years of age. (**b**) The left femur FD nail still maintained good telescoping without implant failure 6 years after surgery.

**Table 1 jpm-12-01151-t001:** Patient demographics and mode of nail failures. OI; Osteogenesis Imperfecta.

ID	Sex	Type of OI	Affected Side	Age (y/o)	Size of Nail (mm)	Number of Failure	Timing of Revision (yr.)	Failure Mode
1	Female	III	Right femur	11	6.4	0		
2	Female	III	Right tibia	4^+4^	4.0	2	2.4	Migration of male component
							3.25	Migration of male component
			Left tibia	5^+9^	4.0	0		
			Right femur	7	4.8	1	3.25	Refracture
3	Male	III	Right femur	1^+6^	3.2	3	2.4	Migration of female component
								Nail bending
							1.3	Migration of female component
							1.4	Delay union
			Left femur	2^+6^	4.8	2	2.75	Migration of female component
							1.4	Refracture
			Right tibia	5	4.8	0		
			Left tibia	5	4.8	1	0.8	Delay union
4	Male	III	Left femur	4^+2^	4.0	0		
			Right femur	5	4.8	1	2.9	Migration of male component
			Left tibia	4^+2^	3.2	1	4.25	Migration of male component
			Right tibia	5	4.0	1	3.3	Delay union
5	Male	III	Left tibia	3	3.2	0		
			Right tibia	4^+6^	4.0	1	2.4	Delay union
								Migration of male component
6	Male	IV	Left femur	11	6.4	1	1.8	Nail bending
			Right femur	11	6.4	0		
7	Female	III	Left femur	5	3.2	1	2.75	Nail bending
			Right femur	4^+8^	3.2	0		
			Left tibia	5	3.2	0		
			Right tibia	4^+8^	3.2	0		

**Table 2 jpm-12-01151-t002:** Failure mode of FD nail.

	Femur (N = 10)	Tibia (N = 10)
Migration of male component	1	4
Migration of female component	3	0
Nail bending	3	0
Refracture on the bone	2	0
Delay union	0	3

## Data Availability

All of the data are registered at MacKay Memorial Hospital, Taipei, Taiwan.

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
