# Peer review of "Surgical Strategy to Decrease the Revision Rate of Fassier–Duval Nailing in the Lower Limbs of Osteogenesis Imperfecta"

_jpm, 2022, doi:10.3390/jpm12071151_

Round 1

Reviewer 1 Report

The article is very interesting and describes a small but significant case series of patients treated are in the long bones of the lower limb.

It does not require complex statistical analysis, the concepts are clearly expressed and are useful, and the iconographic documentation is excellent. 

The type of metal of which the nails are made should be specified.

Recently, the application in the pediatric population of intramedullary synthesis systems already proposed in the adult has been described. Some authors recently described an intramedullary synthesis system in a pediatric population, and they stressed the same concept that you report since page 4: importance of the entry point, good centrodiaphyseal positioning, and fitting in the canal.

Therefore, subsequent to the sentence, "Therefore, we suggest that not only the position of the nail, but also the depth of the purchase and the overall mechanical axis alignment are important, along with a central nail position and bowing correction.", I suggest adding this sentence, "These thoughts have recently been explored in the upper extremity in non-telescopic syntheses" and cite the following articles: (10.1007/s00590-020-02698-z; 10.4103/ajps.AJPS_178_20)

Author Response

The suggestion from reviewer 1: adding the sentence, "These thoughts have recently been explored in the upper extremity in non-telescopic syntheses" and cite the following articles: (10.1007/s00590-020-02698-z; 10.4103/ajps.AJPS_178_20)

We highly appreciated the suggestion. The aim of the proposed study is to investigate the effectiveness of Epibloc system fixation in a general pediatric population suffering from displaced extraphyseal distal radius fractures, instead of osteogenesis imperfecta patients. It emphasis more on the functional outcome after fracture treatment. We will cite the article with a more suitable subject about general population treatment. Thanks for all suggestions!!

Reviewer 2 Report

1.     Can you please elaborate the usage of bisphosphonate? Is there any adverse effect that you have encountered during use?

2.     What are the difficulties when you tried to place the nail in the center of the epiphysis on both anteroposterior and lateral? What is your solution?

3.     What is the latest revision rate in you center?

Author Response

  1. Can you please elaborate the usage of bisphosphonate? Is there any adverse effect that you have encountered during use?

All patients received regular intravenous bisphosphonate treatment with pamidronate disodium at a dose of 30 mg/m2, once every month. The treatment was stopped after surgery but resumed after 3 months post-operatively. Pamidronate is a bisphosphonate, and it has been shown to increase bone mineral density (BMD) and decrease fracture rate, thereby substantially improving functional status. During the first pamidronate infusion, a few patients had low-grade fever. These patients received acetaminophen prior to subsequent pamidronate infusions, and the fever did not recur. No other associated adverse effects were observed.

  1. What are the difficulties when you tried to place the nail in the center of the epiphysis on both anteroposterior and lateral? What is your solution?

This is a key issue over the whole study. Although we have the conclusion that the nail must be placed in the center on both anteroposterior and lateral view, we all know that it is very difficult in practice, especially for small children. First, c-arm positioning intra-operatively cannot be over-emphasized to obtain the true AP and true lateral view of the affected limb. We prefer using the supine position or fracture table during both tibia and femora interventions, thus changing the projection radiation instead of changing the limb position. It is important to try to identify the entry point as precisely as possible. This is especially difficult for proximal femur entry because of the non-ossified small greater trochanter in young children under a non-open and percutaneous approach. The open approach is a possible solution; however, more tissue damage is a concern. Thus, we still use the percutaneous approach under fluoroscopy and accept the possible deviation rather than risk possible damage from the open approach. After a straight and rigid guide wire or reamer has been inserted from the correct entry point, the wire will eventually meet the area of deformity or CORA. We use a multiple percutaneous drilling technique for osteoclasis at the CORA and straighten the limb as the guide wire passes through, which realigns the deformed bone and minimizes surgical trauma.

  1. What is the latest revision rate in you center?

The latest revision rate is 55% in our center till the time of this article submission, as mentioned in the article. Our study is more observational instead of a cohort one. No further revision was required recently after the last surgeries, but longer follow-up time was warranted. We will continue this study to obtain more statistical evident for this cohort and surgical strategies.